# Elastic Alfven waves in elastic turbulence

Atul Varshney [1,2] & Victor Steinberg [1,3]

Speed of sound waves in gases and liquids are governed by the compressibility of the medium. There exists another type of non-dispersive wave where the wave speed depends on stress instead of elasticity of the medium. A well-known example is the Alfven wave, which propagates through plasma permeated by a magnetic field with the speed determined by magnetic tension. An elastic analogue of Alfven waves has been predicted in a flow of dilute polymer solution where the elastic stress of the stretching polymers determines the elastic wave speed. Here we present quantitative evidence of elastic Alfven waves in elastic turbulence of a viscoelastic creeping flow between two obstacles in channel flow. The key finding in the experimental proof is a nonlinear dependence of the elastic wave speed $c_{el}$ on the Weissenberg number Wi, which deviates from predictions based on a model of linear polymer elasticity.

[1] Department of Physics of Complex Systems, Weizmann Institute of Science, 76100 Rehovot, Israel. [2] Institute of Science and Technology Austria, Am Campus 1, 3400 Klosterneuburg, Austria. [3] The Racah Institute of Physics, Hebrew University of Jerusalem, 91904 Jerusalem, Israel. Correspondence and requests for materials should be addressed to A.V. (email: atul.varshney@ist.ac.at) or to V.S. (email: victor.steinberg@weizmann.ac.il)

A small addition of long-chain, flexible, polymer molecules strongly affects both laminar and turbulent flows of Newtonian fluid. In the former case, elastic instabilities and elastic turbulence (ET)[1–5] are observed at Reynolds number Re ≪ 1 and Weissenberg number Wi ≫ 1, whereas in the latter, turbulent drag reduction (TDR) at Re ≫ 1 and Wi ≫ 1 has been found about 70 years ago but its mechanism is still under active investigation[6]. Here both $Re = \rho U D/\eta$ and $Wi = \lambda U/D$ are defined via the mean fluid speed $U$ and the vessel size $D$, and $\rho$, $\eta$ are the density and the dynamic viscosity of the fluid, respectively, and $\lambda$ is the longest polymer relaxation time. ET is a chaotic, inertialess flow driven solely by nonlinear elastic stress generated by polymers stretched by the flow, which is strongly modified by a feedback reaction of elastic stresses[7]. The only theory of ET based on a model of polymers with linear elasticity predicts elastic waves that are strongly attenuated in ET, but elastic waves may play a key role in modifying velocity power spectra in TDR[7,8]. Using the Navier-Stokes equation and the equation for the elastic stresses in uniaxial form of the stress tensor approximation, one can write the polymer hydrodynamic equations in the form of the magneto-hydrodynamic (MHD) equations[8]. Then, by analogy with the Alfven waves in MHD[9,10], one gets the elastic wave linear dispersion relation as $\omega = (\mathbf{k} \cdot \hat{n}) \left[ tr\left( \sigma_{ij} \right)/\rho \right]^{1/2}$ with the elastic wave speed[7,8] $c_{el} = [tr(\sigma_{ij})/\rho]^{1/2}$, where $\omega$ and $\mathbf{k}$ are frequency and wavevector, respectively, $\sigma_{ij}$ is the elastic stress tensor, and $\hat{n}$ is the major stretching direction, similar to the director in nematics. Such an evident difference between the elastic stress tensor characterized by the director and the magnetic field that is the vector, however, does not alter the similarity between the elastic and Alfven waves, since only uniaxial stretching independent of a certain direction is a necessary condition for the wave propagation determined by the stress value[7].

A simple physical explanation of both the Alfven and elastic waves can be drawn from an analogy of the response of either magnetic or elastic tension on transverse perturbations and an elastic string when plucked. As in the case of elastic string, the director is sufficient to define the alignment of the stress. Thus, to excite either Alfven or elastic waves the perturbations should be transverse to the propagation direction, unlike longitudinal sound waves in plasma, gas, and fluid media[11]. The detection of the elastic waves is of great importance for a further understanding of ET mechanism and TDR, where turbulent velocity power spectra get modified according to ref. [7]. Moreover, $c_{el}$ provides unique information about the elastic stresses, whereas the wave amplitude is proportional to the transversal perturbations, both of which are experimentally unavailable otherwise[8].

Numerical simulations of a two-dimensional Kolmogorov flow of a viscoelastic fluid with periodic boundary conditions reveal filamented patterns in both velocity and stress fields of ET[12]. These patterns propagate along the mean flow direction in a wavy manner with a speed $c_{el} \simeq U/2$, nearly independent of Wi. In subsequent studies, extensive three-dimensional Lagrangian

simulations of a viscoelastic flow in a wall-bounded channel with a closely spaced array of obstacles show transition to a time-dependent flow, which resembles the elastic waves[13]. Further, the elastic stress field around the obstacles demonstrates similar traveling filamental structures[12,13] in ET, interpreted as elastic waves[7,8]. However, in both studies neither the linear dispersion relation nor the dependence of wave speed $c_{el}$ on elastic stress–primary signatures of the elastic waves–were examined. Moreover, $c_{el}$ was found to be close to the flow velocity, contradicting the theory[7,8]. Strikingly, an indication of the elastic waves, in numerical studies, originates from observed frequency peaks in the velocity power spectra above the elastic instability[12,13]. Analogous frequency peaks in the power spectra of velocity and absolute pressure fluctuations above the instability were also reported in experiments of a wall-bounded channel flow in a creeping viscoelastic fluid, obstructed by either a periodic array of obstacles[14] or two widely-spaced cylinders[15,16]. These observations were in agreement with numerical simulations[17] and were associated with noisy cross-stream oscillations of a pair of vortices engendered due to breaking of time-reversal symmetry.

Our early attempts to excite the elastic waves both in a curvilinear flow and in an elongation flow of polymer solutions at Re ≪ 1 were unsuccessful[18]. In the ET regime of the curvilinear channel flow, either an excitation amplitude was insufficient and/or an excitation frequency was too high. The reason we chose the elongation flow, realized in a cross-slot micro-fluidic device, is a strong polymer stretching in a well-defined direction along the flow. However, the elongation flow generated in the cross-slot geometry has the highest elastic stresses in a central vertical plane parallel to the flow in the outlet channels–analogous to a stretched vertical elastic membrane. The transverse periodic perturbations in the experiment were applied in a cross-stream direction from the top wall[18], however a more effective method would be to perturb it in a span-wise direction that was difficult to realize in a micro-channel. A higher frequency range of perturbations, compared to that found in the current experiment, was used that lead to the wave excitation with wave numbers in the range of high dissipation.

Here we report evidence of elastic waves observed in elastic turbulence of a dilute polymer solution flow in a wake between two widely-spaced obstacles, hindering a channel flow. The central finding in the experimental proof of the elastic wave observation is a power-law dependence of $c_{el}$ on Wi, which deviates from the prediction based on a model of linear polymer elasticity[7]. The distinctive feature of the current flow geometry is a two-dimensional nature of the ET flow, in the mid-plane of the device, in contrast to other flow geometries studied earlier.

## Results

**Flow structure and elastic turbulence.** The schematic of the experimental setup is shown in Fig. 1, where two-widely spaced obstacles hinder the channel flow of a dilute polymer solution (see

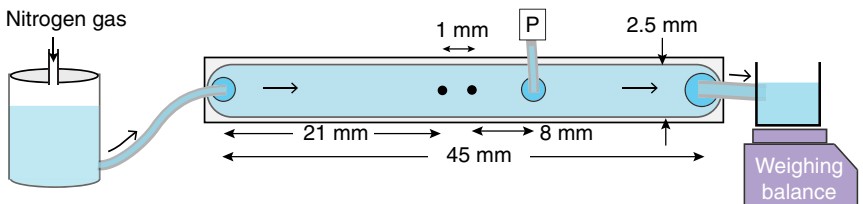

**Fig. 1** Schematic of experimental setup. It consists of a linear channel of dimension $L \times w \times h = 45 \times 2.5 \times 1 \, mm^3$ with two cylindrical obstacles (shown as two black dots), diameter $2R = 0.3 \, mm$ and separated by a distance between the obstacles centers $e = 1 \, mm$, embedded at the center line of the channel. The polymer solution is driven by Nitrogen gas and injected through the inlet into the channel. The fluid exiting the channel outlet is weighed instantaneously as a function of time. An absolute pressure sensor, marked as $P$, after the downstream cylinder is employed to detect pressure fluctuations

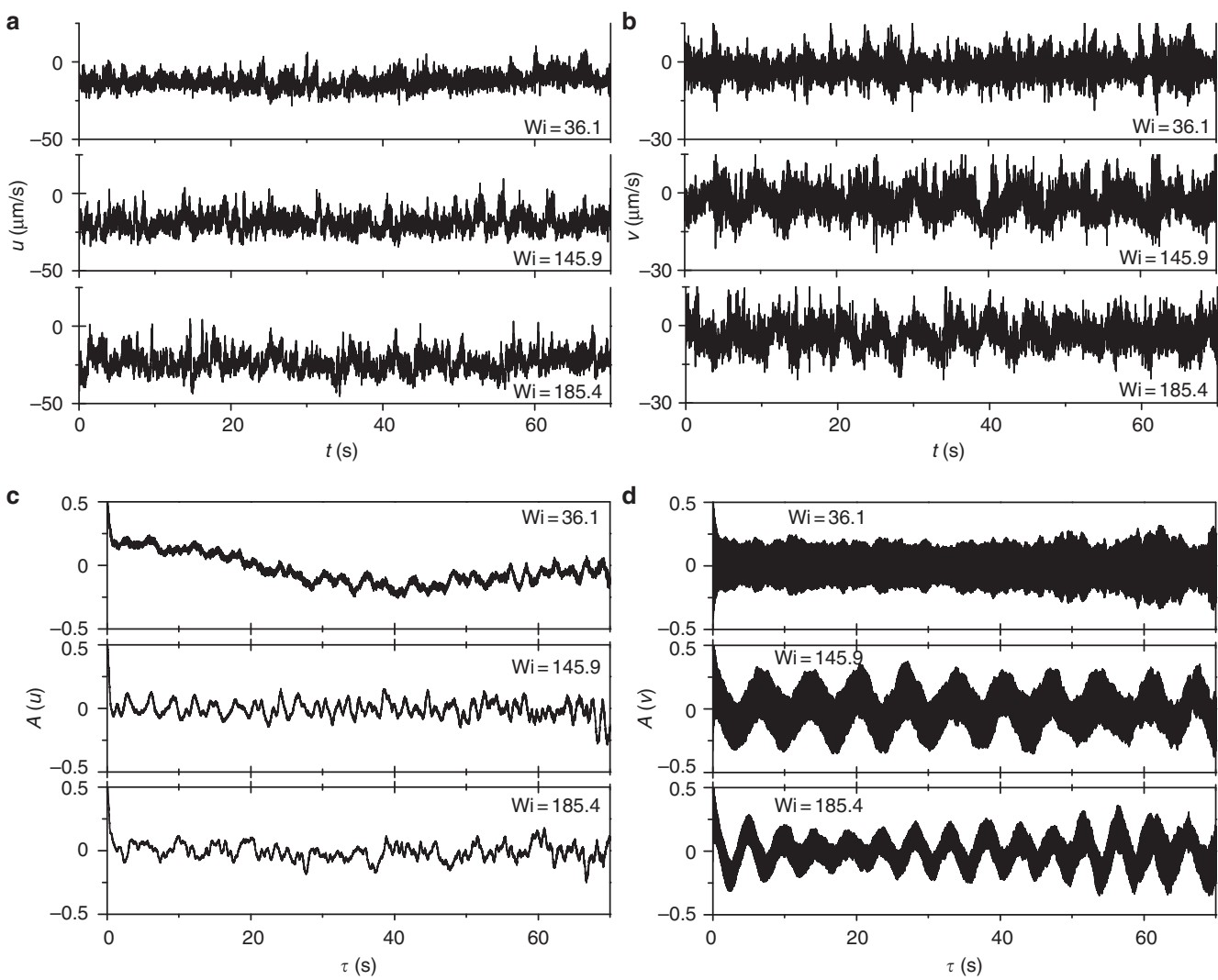

**Fig. 2** Streamwise and cross-stream components of velocity and corresponding autocorrelation functions. Time series of **a** streamwise velocity $u$ and **b** cross-stream velocity $v$, obtained at $(x/R, y/R) = (2.3, 0.03)$, corresponding to the location near the line connecting the centers' of obstacles and close to the center region between the obstacles, for three values of Wi. **c**, **d** Their respective temporal autocorrelation functions $A(u)$ and $A(v)$

Methods section for the experimental setup, solution preparation and its characterization). The main feature of the flow geometry used is the occurrence of a pair of quasi-two-dimensional counter-rotating elongated vortices, in the region between the obstacles, as a result of the elastic instability[15] at Re ≪ 1 and Wi > 1; Re = $2R\bar{u}\rho/\eta$ and Wi = $\lambda\bar{u}/2R$, where obstacles' diameter $2R$ and average flow speed $\bar{u}$ are defined in Methods section. The frequency power spectra of cross-stream velocity $v$ fluctuations show oscillatory peaks at low frequencies[15,16] below $\lambda^{-1}$. Above the elastic instability, the main peak frequency $f_p$ grows linearly with Wi, characteristic to the Hopf bifurcation[15]. The two vortices form two mixing layers with a non-uniform shear velocity profile and with further increase of Wi their dynamics become chaotic, exhibiting ET properties, with vigorous perturbations that intermittently destroy vortices[16] and seemingly excite the elastic waves. The ET flow in the region between the obstacles is shown through long-exposure particle streaks imaging in Supplementary Movies 1–3 for three different Wi.

**Characterization of low frequency oscillations**. To investigate the nature of these oscillations we present time series of the streamwise $u(t)$ and cross-stream $v(t)$ velocity components and

their temporal auto-correlation functions $A(u) = \langle u(t)u(t+\tau)\rangle_t / \langle |u(t)|^2\rangle_t$ and $A(v) = \langle v(t)v(t+\tau)\rangle_t / \langle |v(t)|^2\rangle_t$ in Fig. 2a–d. Distinct oscillations in $v(t)$ contrary to weak noisy oscillations in $u(t)$ indicate flow anisotropy. Further, the cross-stream velocity power spectra $S_f(v)$ as a function of normalized frequency $\lambda f$ for five Wi values in the ET regime are shown in log-lin and log-log coordinates in Fig. 3a, b, respectively. The power spectra $S_f(v)$ exhibit the oscillation peaks at low frequencies up to $\lambda f \leq 40$ with an exponential decay of the peak values (Fig. 3a). These low frequency oscillations look much more pronounced on a linear scale (Supplementary Fig. 1a). Further, these oscillations are also observed in the power spectra of pressure fluctuations $S(P)$ versus $\lambda f$, though not so regular (Supplementary Fig. 1b). The exponential decay of $S_f(v)$ at $\lambda f \leq 40$ implies that only a single frequency (or time) scale is identified for each Wi (Fig. 3a). This frequency $f_d$, for each Wi, is obtained by an exponential fit to the data, i.e., $S_f(v) \sim \exp(-f/f_d)$. The variation of $f_d$ with Wi is shown in the inset in Fig. 3b; it varies from 0.7 to 2.5 Hz in the range of Wi from 75 to 200, which is comparable to oscillation peak frequency $f_p$ (Fig. 4) and larger than $\lambda^{-1}$. Strikingly, on normalization of $f$ with $f_d$ for each Wi, $S_f(v)$ for all Wi collapse on to each other (Fig. 3b). At higher frequencies up to $\lambda f \leq 100$, $S_f(v)$ decay as

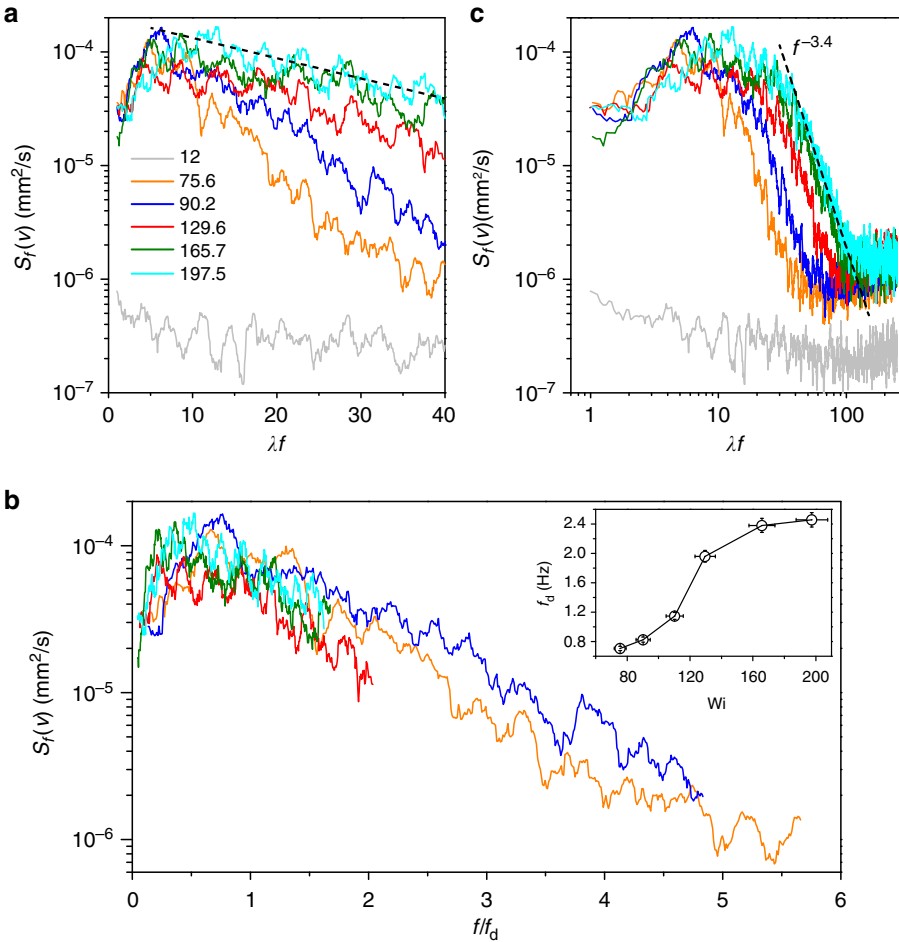

**Fig. 3** Cross-stream velocity power spectra versus normalized frequency in elastic turbulence. **a** Cross-stream velocity power spectra $S_f(v)$ in log-lin coordinates to emphasize an exponential decay of the oscillation peak values at low frequencies $\lambda f \leq 40$. An exponential decay is shown by the dashed line, e.g., for the case of Wi = 197.5. **b** $S_f(v)$ for different Wi collapse on to each other upon normalization of $f$ with $f_d$. Inset: variation of $f_d$ with Wi. The error bars on $f_d$ are estimated based on standard deviation (s.d.) of exponential fit of $S_f(v)$ versus $f$, and for Wi they are calculated based on the s.d. from the mean value of fluid discharge rate $Q$ (see Methods section). **c** $S_f(v)$ in log-log coordinates, for different Wi, to demonstrate the power-law decay at high frequencies ~10 < $\lambda f \leq 100$. The spectra are obtained at $(x/R, y/R) = (5.2, 0.56)$, which is close to the downstream obstacle and to the center of the upper large vortex. The dashed line in **c** is a fit to the data at high frequencies with the power-law exponent $\alpha_f \simeq -3.4 \pm 0.1$, typical to the ET regime. $S_f(v)$ of steady flow is shown by gray lines in **a**, **c**

the power-law with the exponent $\alpha_f = -3.4 \pm 0.1$ typical for ET[5] (Fig. 3c). Contrary to a general case, where the power-law decay of $S_f(v)$ corresponding to ET[3–5] commences at $\lambda f \approx 1$, the low frequency oscillations cause the power-law spectra start to decay at higher frequencies $10 < \lambda f < 40$, perhaps due to an additional mechanism of energy pumping into ET associated with the low frequency oscillations. In addition, $S(P)$ exhibit the power spectra decay in the high frequency range $10 < \lambda f < 100$ with the exponent close to $-3$ (see the bottom inset in Fig. 2 in ref. [16]), characteristic to the ET regime[19]. It should be emphasized that the power spectra of the streamwise velocity $S_f(u)$ do not show the low frequency oscillations and decays with a power-law exponent $\alpha \leq 2$.

Figure 4 shows the dependence of $f_p$ in a wide range of Wi. The first elastic instability, characterized as the Hopf bifurcation, occurs at low Wi, where $f_p$ grows linearly with Wi–in accord with our early results[15]. At higher Wi in the ET regime, $f_p$(Wi) dependence becomes nonlinear at Wi ≥ 60. In the inset in Fig. 4, we present the same data for $f_p$ as a function of $Wi_{int}$. Here, the Weissenberg number of the inter-obstacle velocity field is defined as $Wi_{int} = \lambda \dot\gamma$ and $\dot\gamma (= \langle \partial u/\partial y \rangle_t)$ is the time-averaged shear-rate in the cross-stream direction in the inter-obstacle flow region. The parameter $Wi_{int}$ is relevant to the description of elastic waves

in ET flow between the obstacles' region. The inset in Fig. 5b shows a linear dependence of $Wi_{int}$ on Wi.

**Dependence of elastic wave speed on $Wi_{int}$.** Figure 5a shows a family of temporal cross-correlation functions $C_v(\Delta x, \tau) = \langle v(x, t) v(x + \Delta x, t + \tau) \rangle_t / \langle v(x, t) v(x + \Delta x, t) \rangle_t$ of $v$ between two spatially separated points, with their distance being $\Delta x$, located on a horizontal line at $y/R = 0.18$ for Wi = 148.4. A gaussian fit to $C_v(\Delta x, \tau)$ in the vicinity of $\tau = 0$ yields the peak value $\tau_p$ at a given $\Delta x$. A linear dependence of $\Delta x$ on $\tau_p$ (e.g., Fig. 5a inset for Wi = 148.4) provides the perturbation propagation velocity as $c_{el} = \Delta x/\tau_p$. The variation of $c_{el}$ as a function of $Wi_{int}$ is presented in Fig. 5b together with nonlinear fit of the form $c_{el} = A(Wi_{int} - Wi_{int}^c)^\beta$, where $A = 8.9 \pm 1.2$ mm s$^{-1}$, $\beta = 0.73 \pm 0.12$, and onset value $Wi_{int}^c = 1.75 \pm 0.2$. The same data of $c_{el}$ is plotted against Wi (see Supplementary Fig. 3) and fitted as $c_{el} \sim (Wi - Wi_c)^\beta$ that yields the onset value $Wi_c = 59.7 \pm 1.8$.

**Discussion**

In the light of the predictions[7], it is surprising to observe the elastic waves in the ET regime due to their anticipated strong

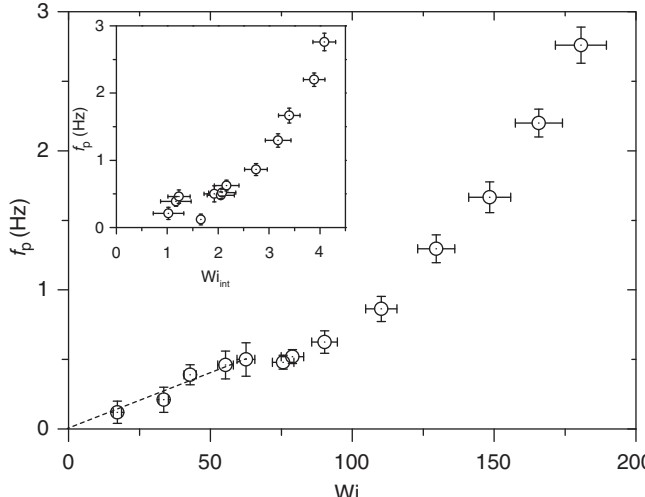

**Fig. 4** Dependence of oscillation peak frequency on Wi. Dashed line is a linear fit to the data in the regime above the elastic instability. Inset: $f_p$ as a function of $Wi_{int}$. The error bars on $f_p$ are estimated from the spectral width of the oscillatory peaks of $S_f(v)$, and for $Wi_{int}$ they are calculated based on the s.d. from the mean value of $(\partial u/\partial y)$

attenuation. An estimate of the wave number $k = \omega/c_{el} = 2\pi f_p/c_{el}$ from $c_{el}$ (Fig. 5b) and $f_p$ (Fig. 4) provides $k$ in the range between 0.63 and 1.3 mm$^{-1}$ (Supplementary Fig. 2). The corresponding wavelengths ($\sim 2\pi/k$) are significantly larger than the inter-obstacle spacing $e - 2R = 0.7$ mm. The spatial velocity power spectra $S_k$ is limited by a size of the observation window of about 0.7 mm that gives $k_x \approx 9$ mm$^{-1}$, much larger than the wave numbers calculated above. Thus, the low $k_x$ part of $S_k(v)$, where the elastic wave peaks can be anticipated, is not resolved by the spatial velocity spectra (Supplementary Fig. 4b). The power-law decay with $\alpha_k \approx -3.3$ is found at low $k_x$ followed by a bottleneck part and a consequent gradual power-law decay with an exponent $\sim -0.5$ at higher $k_x$ (Supplementary Fig. 4b), unlike $S_f(v)$, where the peaks appear at low $f$ and the steep power-law decay with the exponent $\alpha_f = -3.4$ at higher $f$ (see Fig. 3b). The spatial stream-wise velocity power spectra $S_k(u)$, obtained at the same Wi and near the center line $y/R = 0.01$, are similar to $S_k(v)$ at low $k_x$ and decays gradually with exponent $\sim -0.3$ at higher $k_x$ (Supplementary Fig. 4a).

The observed nonlinear dependence of $c_{el}$ on $Wi_{int}$ differs from the theoretical prediction based on the Oldroyd-B model[7,8]. The expression for the elastic wave speed in the model[20] gives $c_{el} = [tr(\sigma_{ij})/\rho]^{1/2} \approx (N_1/\rho)^{1/2}$, where $N_1 = 2Wi_{int}^2\eta/\lambda$ is the first normal stress difference. Then one obtains $c_{el} = (2\eta/\rho\lambda)^{1/2}Wi_{int}$. First, $c_{el}$ is proportional to $Wi_{int}$ and second, the coefficient in the expression for the parameters used in the experiment is estimated to be $(2\eta/\rho\lambda)^{1/2} = 4.5$ mm s$^{-1}$. Taking into account that the model[7,8] and the estimate of elastic stress are based on linear polymer elasticity[20], whereas in experiments polymers in ET flow are stretched far beyond the linear limit[21], thus it is not surprising to find the quantitative discrepancies between them. Indeed, the value of the coefficient found from the fit (8.9 mm s$^{-1}$) and estimated theoretical value (4.5 mm s$^{-1}$) differ almost by a factor of two (see Fig. 5b). Moreover, for the maximal value of $c_{el} = 17$ mm s$^{-1}$ (at $Wi_{int} \approx 4$) obtained in the experiment, an estimate of elastic stress gives $\langle\sigma\rangle = c_{el}^2\rho = 0.37$ Pa that is lower but comparable with $\langle\sigma\rangle \approx 1$ Pa obtained from the experiment on stretching of a single polymer T4DNA molecule at similar concentrations[21]. Thus, both the $c_{el}$ dependence on $Wi_{int}$ and the coefficient value indicate that the Oldroyd-B model based on linear polymer elasticity cannot quantitatively describe the elastic

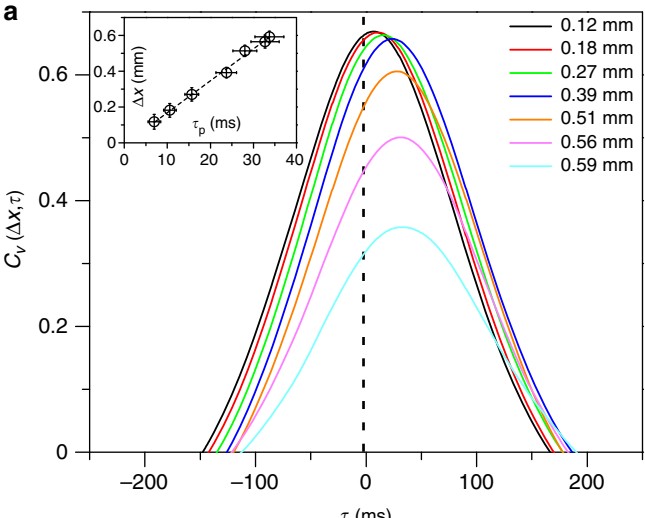

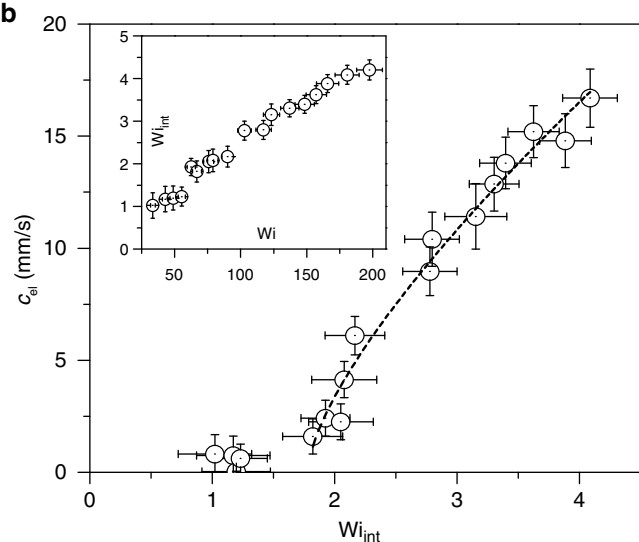

**Fig. 5** Elastic wave speed versus $Wi_{int}$. **a** Cross-correlation functions of the cross-stream velocity $C_v(\Delta x, \tau)$ versus lag time $\tau$ for different values of $\Delta x$, obtained at $y/R = 0.18$ and for $Wi = 148.4$. Inset: $\Delta x$ versus $\tau_p$ for $Wi = 148.4$, and a slope of linear fit to it (shown by dashed line) provides $c_{el}$. The error bars on $\Delta x$ are determined by the spatial resolution of measurements, and for $\tau_p$ they are estimated based on the s.d. of gaussian fit of $C_v(\Delta x, \tau)$. **b** Dependence of $c_{el}$ on $Wi_{int}$, where the dashed line is a fit of the form $c_{el} = A(Wi_{int} - Wi_{int}^c)^\beta$, where $A = 8.9 \pm 1.2$ mm s$^{-1}$, $\beta = 0.73 \pm 0.12$, and onset value $Wi_{int}^c = 1.75 \pm 0.2$. Inset: $Wi_{int}$ versus $Wi$. The error on $c_{el}$ is estimated based on the s.d. of the linear fit of $\Delta x$ versus $\tau_p$

wave speed and so the elastic stresses. Another aspect of this result is the Mach number $Ma \equiv \bar{u}/c_{el}$; the maximum value achieved in the experiment is $Ma_{max} = \bar{u}_{max}/c_{el} \approx 0.3$, contrast to what is claimed in refs.[22,23] due to a wrong definition based on the elasticity $El = Wi/Re$ instead of elastic stress $\sigma$ used for the estimation of $c_{el}$ and Ma.

We discuss two possible reasons related to the detection of the elastic waves. As indicated in the introduction, the key feature of the current geometry is a two-dimensional nature of the chaotic flow, at least in the mid-plane of the device (see Fig. 4SM in Supplemental Material of ref. [16]), that makes it analogous to a stretched elastic membrane. This flow structure is different from three-dimensional elastic turbulence in other studied flow geometries and thus may explain the failure in the earlier attempts to observe the elastic waves. Another qualitative discrepancy with

the theory[7,8] is the predicted strong attenuation of the elastic waves in ET. Below we estimate the range of the wave numbers with low attenuation for the elastic waves and compare with the observed values.

There are two mechanisms of the elastic wave attenuation, namely polymer (or elastic stress) relaxation and viscous dissipation[7,8]. The former has scale-independent attenuation $\lambda^{-1}$, which at the weak attenuation satisfies the relation $\omega\lambda > 1$, and the latter provides low attenuation[24] at $\eta k^2/\rho\omega < 1$. The first condition leads to $ks > 1$, where $s = \text{Wi}_{\text{int}}(2\eta\lambda/\rho)^{1/2}$ that provides a minimum wave number in the ET regime as $k_{\text{min}} > s^{-1} = 6.3 \times 10^{-3}$ mm$^{-1}$ for $\text{Wi}_{\text{int}} = 4$. The maximum value $k_{\text{max}}$ follows from the second condition that gives $k\Lambda < 1$ at $\Lambda = (\text{Wi}_{\text{int}})^{-1}(\eta\lambda/2\rho)^{1/2}$. Thus, one obtains in the ET regime $k_{\text{max}} < \Lambda^{-1} = 0.2$ mm$^{-1}$ for $\text{Wi}_{\text{int}} = 4$ and therefore, the range of the wave numbers with the low attenuation is rather broad $6.3 \times 10^{-3} < k < 0.2$ mm$^{-1}$ and lies far outside of the $k$-range of $S_k(u)$ and $S_k(v)$ presented in Supplementary Fig. 4, where the range of the wave numbers of the elastic waves is not resolved. However, the range of the observed wave number $0.63 \leq k \leq 1.3$ mm$^{-1}$ of the elastic waves, shown in Supplementary Fig. 2, is sufficiently close to the estimated upper bound of $k$.

## Methods

**Experimental setup.** The experiments are conducted in a linear channel of $L \times w \times h = 45 \times 2.5 \times 1$ mm$^3$, shown schematically in Fig. 1. The channel is prepared from transparent acrylic glass (PMMA). The fluid flow is hindered by two cylindrical obstacles of $2R = 0.30$ mm made of stainless steel separated by a distance of $e = 1$ mm and embedded at the center of the channel. Thus the geometrical parameters of the device are $2R/w = 0.12$, $h/w = 0.4$ and $e/2R = 3.3$ (see Fig. 1). The longitudinal and transverse coordinates of the channel are $x$ and $y$, respectively, with $(x, y) = (0, 0)$ lies at the center of the upstream cylinder. The fluid is driven by $N_2$ gas at a pressure up to ~10 psi and is injected via an inlet into the channel.

**Preparation and characterization of polymer solution.** As a working fluid, a dilute polymer solution of high molecular weight polyacrylamide (PAAm, $M_w = 18$ MDa; Polysciences) at concentration $c = 80$ ppm ($c/c^* \simeq 0.4$, where $c^* = 200$ ppm is the overlap concentration for the polymer used[25]) is prepared using a water-sucrose solvent with sucrose weight fraction of 60%. The solvent viscosity, $\eta_s$, at 20 °C is measured to be 100 mPa·s in a commercial rheometer (AR-1000; TA Instruments). An addition of the polymer to the solvent increases the solution viscosity, $\eta$, of about 30%. The stress-relaxation method[25] is employed to obtain longest relaxation time ($\lambda$) of the solution and it yields $\lambda = 10 \pm 0.5$ s.

**Flow discharge measurement.** The fluid exiting the channel outlet is weighed instantaneously $W(t)$ as a function of time $t$ by a PC-interfaced balance (BA210S, Sartorius) with a sampling rate of 5 Hz and a resolution of 0.1 mg. The time-averaged fluid discharge rate $\bar{Q}$ is estimated as $\Delta W/\Delta t$. Thus, Weissenberg and Reynolds numbers are defined as $\text{Wi} = \lambda\bar{u}/2R$ and $\text{Re} = 2R\bar{u}\rho/\eta$, respectively; here $\bar{u} = \bar{Q}/\rho wh$ and fluid density $\rho = 1286$ Kg m$^{-3}$.

**Imaging system.** For flow visualization, the solution is seeded with fluorescent particles of diameter 1 μm (Fluoresbrite YG, Polysciences). The region between the obstacles is imaged in the mid-plane via a microscope (Olympus IX70), illuminated uniformly with LED (Luxeon Rebel) at 447.5 nm wavelength, and two CCD cameras attached to the microscope: (i) GX1920 Prosilica with a spatial resolution $1000 \times 500$ pixel at a rate of 65 fps and (ii) a high resolution CCD camera XIMEA MC124CG with a spatial resolution $4000 \times 2200$ pixel at a rate of 35 fps, are used to acquire images with high temporal and spatial resolutions, respectively. We perform micro particle image velocimetry[26] (μPIV) to obtain the spatially-resolved velocity field $\mathbf{U} = (u, v)$ in the region between the cylinders. Interrogation windows of $16 \times 16$ pixel$^2$ ($26 \times 26$ μm$^2$) for high temporal resolution images and $64 \times 64$ pixel$^2$ ($10 \times 10$ μm$^2$) for high spatial resolution images, with 50% overlap are chosen to procure $\mathbf{U}$.

## Data availability
The data that support the findings of this study are available from the corresponding authors upon reasonable request.

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

## Acknowledgements
We thank Guy Han and Yuri Burnishev for technical support. A.V. acknowledges support from the European Union's Horizon 2020 research and innovation programme under the Marie Skłodowska-Curie grant agreement No. 754411. This work was partially supported by the Israel Science Foundation (ISF; grant #882/15) and the Binational USA-Israel Foundation (BSF; grant #2016145).

## Author contributions

A.V. and V.S. designed the experiment. A.V. performed the measurements and together with V.S. analyzed the data. Both authors discussed the results and wrote the manuscript.

## Additional information

**Competing interests:** The authors declare no competing interests.

