## [Peer Review File · Nature Communications]

Reviewer #1 (Remarks to the Author):

The paper reports the first experimental observation of elastic Alfvén waves in elastic turbulence. These waves were predicted fifteen years ago in Ref. 11 (see also Ref. 3 for high-Reynolds number case) however were never observed. The waves' existence was questioned. The work is highly relevant for the development of the area and I gladly recommend the publication after several points are addressed in the revision.

It was stressed in Ref. 11 that the "magnetic field" of polymers is defined up to the sign. It can be parallel or antiparallel to an eigenvector of the stress. (It would help the reader telling that what the authors call "main stress direction" is the eigenvector that belongs to the eigenvalue which is much larger than the rest of the eigenvalues, the stress is uniaxial). Does this introduce a difference in comparison with Alfvén waves in magnetohydrodynamics and could the authors talk of this?

What did the authors do that was not done previously for observing the waves? This would help the reader to see the authors' contribution. I could not see this from the review provided by the authors.

The paper aims at experimental establishment of the waves. Thus it would help if it provided description of when these waves must occur. Why they did not see them in the previous experiment however saw this time (see above)? What is the range of situations in elastic turbulence where the waves would occur?

It seems that considerations of the points above would make the paper of more impact.

with best wishes,
Itzhak Fouxon

Reviewer #2 (Remarks to the Author):

This paper discusses the experimental observation of the analogous of Alfvén waves in an elastic turbulent flow produced in a dilute polymer solution. The main quantitative result is the linear dependence of the wave speed on the Weissenberg number of the flow, in agreement with model prediction.

Elastic turbulence has been discovered recently by one of the Authors of the present paper. It is a new regime of dilute polymer solution which appears at low Reynolds numbers and high Weissenberg numbers. It is interesting both for our general understanding of complex flows (in this case, elastic flow) and for potential application, such as mixing at low Reynolds numbers. In spite of this general remark, I am sorry to say that I cannot recommend this manuscript for publication in Nature Communications.

The paper is very technical, not clearly organized and written, and I had many difficulties to understand what the Authors did. Most of the results presented are very noisy and, in my opinion, do not support the claims made in the paper.

My suggestion is to submit the manuscript to a more technical journal with explicit discussion of the technical issues which are hidden in the present form.

One simple example from page 1:

"Using the Navier-Stokes equation and equation for the elastic stresses in uniaxial form of the stress tensor approximation, one gets the polymer hydrodynamic equation in the form of the magneto-hydrodynamic (MHD) ones."

I am afraid that the reader who is not familiar with viscoelastic models, their reduction to the uniaxial form and MHD equation (and Navier-Stokes equations) will understand very little of the above sentence and, more important, of the results presented in the paper.

Specific points.

1. The setup and the measurements are not clear without looking at the supp materials (and also not very clear there). At least a sketch of the experimental setup should be presented in the main paper, together with a better description of the flow produced, since it seems to be very inhomogeneous.

2. Data shown in Figs. 2 and 4 are very noisy and therefore the claimed exponential/power-law behavior is not convincing. It is not possible to increase the statistics in order to get better spectra?

The exponential fit in Fig.2a seems to be not compatible with the case $Wi=165.7$ (and also with data at lower Wi). Why these data are different?

How the power-law behavior in Fig.2b has been obtained? Is the exponent 3.4 a fit or what? What is the error estimation on this exponent?

Authors should provide more convincing results, e.g. compensated plots or local scaling. What are the scaling exponents for the other Wi numbers?

Similar considerations apply to Fig.4. Probably there are too many curves, and the fit with power-law is not convincing (and again, what is the error on the exponents?).

3. Why some results are presented in terms of Wi and other as a function of Wi_{int} ? I understand that the relation between Wi and Wi_{int} is almost linear, but what is the reason to use two different Weissenberg numbers? What is the physical meaning of Wi_{int} ?

The use of two Wi numbers helps to confuse the reader. Moreover, Wi_{int} is used on page 2, while it is defined on page 3.

4. A point which should be discussed is the role of the flow.

Since Wi is proportional to the fluid speed, and Wi_{int} to Wi , the velocity obtained from the cross-correlation of the cross-stream velocity could be proportional to Wi_{int} simply because of the presence of the flow which transport the perturbation (i.e. c proportional to u). I am sure that the Author have excluded this possibility, but I think that this could be discussed.

5. Minor points:

Why are the Wi numbers in Fig.1 different from those in Fig.2?

Fig.3a: in the inset the slope gives $1/c$, not c (alternatively, the labels are wrong)

November 30, 2018

Reviewer #1 (Remarks to the Author):

The paper reports the first experimental observation of elastic Alfvén waves in elastic turbulence. These waves were predicted fifteen years ago in Ref. 11 (see also Ref. 3 for high-Reynolds number case) however were never observed. The waves' existence was questioned. The work is highly relevant for the development of the area and I gladly recommend the publication after several points are addressed in the revision.

It was stressed in Ref. 11 that the "magnetic field" of polymers is defined up to the sign. It can be parallel or antiparallel to an eigenvector of the stress. (It would help the reader telling that what the authors call "main stress direction" is the eigenvector that belongs to the eigenvalue which is much larger than the rest of the eigenvalues, the stress is uniaxial). Does this introduce a difference in comparison with Alfvén waves in magnetohydrodynamics and could the authors talk of this?

Specifically regarding this comment, we added in the second paragraph in the Introduction a discussion of the difference between the magnetic field vector and the director of the main stress field similar to nematics. Comparison with the stretched string provides a good explanation that one does not need vector field to get elastic waves.

What did the authors do that was not done previously for observing the waves? This would help the reader to see the authors' contribution. I could not see this from the review provided by the authors.

We discuss now in length both in Introduction and in conclusion all possible reasons for failures in the previous attempts to detect the elastic waves and compare with the current setup.

The paper aims at experimental establishment of the waves. Thus it would help if it provided description of when these waves must occur. Why they did not see them in the previous experiment however saw this time (see above)? What is the range of situations in elastic turbulence where the waves would occur?

In our revised version we added a paragraph in the Introduction section to address the points raised by the referee. The text reads:

"Our early attempts to excite the elastic waves both in a curvilinear flow and in an elongation flow of polymer solutions at $Re \ll 1$ were unsuccessful¹⁹. In the ET regime of the curvilinear channel flow, either an excitation amplitude was insufficient or/and an excitation frequency was too high. And in the elongation flow, realized in a cross-slot micro-fluidic device, the attempt to observe the elastic waves was also failed, probably due to a higher frequency range of perturbations used in the experiment compared to that found in the current experiment. The choice of the perturbation frequencies was then based on the theoretical estimates of the elastic stress, which was underestimated due to use of a linear polymer model²⁰. Consequently, the frequencies used in the earlier experiments were located inside the range of high dissipation. It is worth pointing out that there is no theory that considers the possibility to observe the elastic waves in such flow geometry. Moreover, another possible reason that elastic waves were undetected is the structure of the elongation flow itself. Indeed, the elongation flow generated in the cross-slot geometry has the highest elastic stresses in a central vertical plane of the outlet channels—analogue to a stretched

November 30, 2018

vertical elastic membrane. Then an effective way to excite the membrane is to perturb it in a span-wise direction instead of cross-stream direction used in the experiment¹⁹.”

and in the following paragraph

“Further, we discuss two possible reasons related to the detection of the elastic waves. As indicated in the introduction, the key feature of the current geometry is a two-dimensional nature of the chaotic flow, at least in the mid-plane of the device (see Fig. 4SM in Supplemental Material of Ref.¹⁷), that makes it analogous to a stretched elastic membrane. This flow structure is different from three-dimensional elastic turbulence in other studied flow geometries and thus may explain the failure in the earlier attempts to observe the elastic waves. Another evident qualitative discrepancy with the theory^{3,11} is the predicted strong attenuation of the elastic waves in ET. Below we estimate the range of the wave numbers with low attenuation for the elastic waves and compare with the observed values.”

Reviewer #2 (Remarks to the Author):

This paper discusses the experimental observation of the analogous of Alfvén waves in an elastic turbulent flow produced in a dilute polymer solution. The main quantitative result is the linear dependence of the wave speed on the Weissenberg number of the flow, in agreement with model prediction.

Elastic turbulence has been discovered recently by one of the Authors of the present paper. It is a new regime of dilute polymer solution which appears at low Reynolds numbers and high Weissenberg numbers. It is interesting both for our general understanding of complex flows (in this case, elastic flow) and for potential application, such as mixing at low Reynolds numbers.

In spite of this general remark, I am sorry to say that I cannot recommend this manuscript for publication in Nature Communications.

The paper is very technical, not clearly organized and written, and I had many difficulties to understand what the Authors did. Most of the results presented are very noisy and, in my opinion, do not support the claims made in the paper.

My suggestion is to submit the manuscript to a more technical journal with explicit discussion of the technical issues which are hidden in the present form.

The main breakthrough reported in the paper is the observation of the elastic waves which speed is strongly depends on Wi . It signifies the dependence of the elastic wave speed on the elastic stress, the key feature of these waves. Taking into account the comment of the Referee about not convincing linear fit, we made more detailed analysis by using nonlinear fit of c_{el} vs Wi_{int} and found out that the quality of this fit is better than linear. We correspondingly revised the text.

One simple example from page 1:

"Using the Navier-Stokes equation and equation for the elastic stresses in uniaxial form of the stress tensor approximation, one gets the polymer hydrodynamic equation in the form of the magneto-hydrodynamic (MHD) ones."

I am afraid that the reader who is not familiar with viscoelastic models, their reduction to the uniaxial form and MHD equation (and Navier-Stokes equations) will understand very little of the above sentence and, more important, of the results presented in the paper.

The linear elastic model that predict the elastic waves in polymer solutions and its similarity with the Magneto-hydrodynamic equation are delineated in Refs. [3, 11]. Therefore, we do not discuss in details the same instead we emphasize more on the dependency of elastic wave speed on Weissenberg number.

Specific points.

1. The setup and the measurements are not clear without looking at the supp materials (and also not very clear there). At least a sketch of the experimental setup should be presented in the main paper, together with a better description of the flow produced, since it seems to be very inhomogeneous.

We moved experimental setup figure from supplementary to main text as Fig. 1. The setup and method to produce flow are described in Materials and methods section of the manuscript.

Moreover, we show ET flow between the obstacles' region through long-exposure particle streaks in Supplementary movies SM1-3, obtained at three different Wi .

2. Data shown in Figs. 2 and 4 are very noisy and therefore the claimed exponential/power-law behavior is not convincing. It is not possible to increase the statistics in order to get better spectra?

These are real cross-stream oscillations as a result of ET flow, and not the noise that referee misunderstood. Further, the signal level for steady flow at low Weissenberg number is shown by grey lines in Fig 2 (Fig. 3 in present version) and the level in regime of elastic turbulent flow is several orders of magnitude higher. To give a better impression of the flow, we show three supplementary movies for three Weissenberg numbers that will perhaps ease understanding the appearance of low frequency oscillations coming from flow. In addition, we show dependency of oscillation peak frequency on Wi (or Wi_{int}) as Fig. 4 in the manuscript. We moved the spatial spectra S_k to Supplementary Fig. S4 since the spectra is limited by a size of observation window (0.7 mm) and cannot resolve the low frequency spatial oscillations anticipated at $k_x \sim 0.63 \text{ mm}^{-1}$ ($l \sim 10 \text{ mm}$). Nevertheless, the spectra show -3.3 scaling at low k_x which is typical to the ET flow.

The exponential fit in Fig.2a seems to be not compatible with the case $Wi=165.7$ (and also with data at lower Wi). Why these data are different?

The dashed line in Fig 2a (now Fig. 3a) is an exponential fit to the case of $Wi=197.5$. In our revised version, we fit exponential to $S_v(v)$, for each Wi , as $S_v(v) \sim \exp(-v/f_d)$ and obtain f_d as a function of Wi (shown in the inset of Fig. 3b). Further, we normalize v with f_d as (v/f_d) and plotted with $S_v(v)$ that results in a collapse of $S_v(v)$ for all Wi in ET (see Fig. 3b)—highlighting that f_d is a characteristic decay frequency (or inverse time) scale.

How the power-law behavior in Fig.2b has been obtained? Is the exponent 3.4 a fit or what? What is the error estimation on this exponent?

In ET regime, the frequency power spectra $S_v(v)$ for each Wi is fitted with power-law that yields exponent -3.4 ± 0.1 . The error on the exponent is now specified in the present version of the manuscript.

Authors should provide more convincing results, e.g. compensated plots or local scaling. What are the scaling exponents for the other Wi numbers?

The value of the scaling exponent of the velocity power spectra in elastic turbulence at $Re \ll 1$ was discussed in literature in length for past 15 years. The physical mechanism of the power-law decay in ET is different from the well-known energy cascade in the inertial range of scales in hydrodynamic turbulence at $Re \gg 1$, where well-defined exponent value is predicted and observed. In ET according to theory the exponent value should be steeper than -3 and in experiments and numerical simulations it varies between about -3 and about -4 or even steeper. Hence, the exponent value of -3.4 obtained in present experiments represents well the ET flow. So the compensated plot used in hydrodynamic turbulence to define the range of the inertial range of scales, in a more precise way, is irrelevant in the case of ET. However, we show the compensated plot here in our response in Fig. 1.

Figure 1: Compensated plot of velocity power spectra $v^{\beta} S_v(v)$ versus λv for different Wi in ET. Grey line represents the steady flow.

Similar considerations apply to Fig.4. Probably there are too many curves, and the fit with power-law is not convincing (and again, what is the error on the exponents?).

For both frequency and spatial power spectra, we now show curves obtained only in the ET regime and rest of them are removed. The exponent for S_k spectra is found to be -3.3 ± 0.1 (see Supplementary Fig. S4).

3. Why some results are presented in terms of Wi and other as a function of Wi_{int} ? I understand that the relation between Wi and Wi_{int} is almost linear, but what is the reason to use two different Weissenberg numbers? What is the physical meaning of Wi_{int} ? The use of two Wi numbers helps to confuse the reader.

The difference between Wi and Wi_{int} in the parameters used in the definition: Wi is the global Weissenberg number defined via the average shear rate in a channel flow velocity, which the average velocity in the channel flow divided on the obstacle diameter. This velocity describes a laminar flow in the channel. On the other hand, to determine correctly ET flow in the inter-obstacle region, where the main action takes place, we introduce the relevant “local” Wi_{int} , which is defined via the time-averaged strongly fluctuating velocity gradient responsible for the polymer stretching in the inter-obstacle region (one can use here equally well rms of the velocity gradient which are comparable in value in ET). Since the polymers are mostly stretched in the vicinity of obstacles and in the wake region between the obstacles, therefore it is relevant to use Wi_{int} , as opposed to Wi from bulk flow, for the description of the elastic waves in ET.

Moreover, Wi_{int} is used on page 2, while it is defined on page 3.

We have corrected it in the present manuscript version.

4. A point which should be discussed is the role of the flow.

Since Wi is proportional to the fluid speed, and Wi_{int} to Wi , the velocity obtained from the cross-correlation of the cross-stream velocity could be proportional to Wi_{int} simply because of the presence of the flow which transport the perturbation (i.e. c proportional to u). I am sure that the

November 30, 2018

Author have excluded this possibility, but I think that this could be discussed.

The elastic wave speed is obtained from the cross-correlation of cross-stream component of velocity. The magnitude of obtained elastic wave speed is much larger than the stream-wise and cross-stream components of velocity in the obstacles' region (see Fig. 2), thus these perturbations are not being advected by the mean flow. It's the case analogous to a stretched elastic string and any transverse perturbations propagate with speed defined by the elastic stress of the string. We discuss this point in the present manuscript version.

5. Minor points:

Why are the Wi numbers in Fig.1 different from those in Fig.2?

The data in Fig. 1 (Fig. 2 in the present version) is recorded with high-frame rate camera and for a rather short time for only a few Wi values to exemplify the flow. However, for the better statistics to compute spectra (Fig. 3) etc. we recorded with relatively high-spatial resolution and at moderate frame-rate but for longer times.

Fig.3a: in the inset the slope gives $1/c$, not c (alternatively, the labels are wrong)

We have rectified this error (see Fig. 5a inset) in the present manuscript version.

Reviewer #1 (Remarks to the Author):

The authors have successfully addressed the comments that I had for the previous version. I have only one remark: the director's direction \hat{n} on the first page is the major stretching direction which is different from the direction of the eigenvector of strain with largest eigenvalue (in fact the major stretching direction was reported to be more correlated with vorticity than with the eigenvector). When this is corrected I gladly recommend the publication of this work with no need for further review.

with best wishes,

Itzhak Fouxon

Reviewer #1 (Remarks to the Author):

The authors have successfully addressed the comments that I had for the previous version. I have only one remark: the director's direction \hat{n} on the first page is the major stretching direction which is different from the direction of the eigenvector of strain with largest eigenvalue (in fact the major stretching direction was reported to be more correlated with vorticity than with the eigenvector). When this is corrected I gladly recommend the publication of this work with no need for further review.

We are grateful to the Reviewer for the further remark and recommending the manuscript for publication in Nature Communications. We have incorporated the remark in our revised manuscript.

The text in the Introduction reads:

“Then by analogy with the Alfvén waves in MHD^{9,10}, one gets the elastic wave linear dispersion relation as $\omega = (\mathbf{k} \cdot \hat{n})[\text{tr}(\sigma_{ij})/\rho]^{1/2}$ with the elastic wave speed^{7,8} $c_{el} = [\text{tr}(\sigma_{ij})/\rho]^{1/2}$, where ω and \mathbf{k} are frequency and wavevector respectively, σ_{ij} is the elastic stress tensor, and \hat{n} is the major stretching direction, similar to the director in nematics.”